# SVA Regulation of Transposable Element Clustered Transcription within the Major Histocompatibility Complex Genomic Class II Region of the Parkinson’s Progression Markers Initiative

**DOI:** 10.3390/genes15091185

**Published:** 2024-09-09

**Authors:** Jerzy K. Kulski, Abigail L. Pfaff, Sulev Koks

**Affiliations:** 1Faculty of Health and Medical Sciences, School of Biomedical Science, The University of Western Australia, Crawley, WA 6009, Australia; yurek.kulski@uwa.edu.au; 2Department of Molecular Life Science, Division of Basic Medical Science and Molecular Medicine, Tokai University School of Medicine, Isehara 259-1193, Japan; 3Perron Institute for Neurological and Translational Science, Perth, WA 6009, Australia; abi.pfaff@murdoch.edu.au; 4Centre for Molecular Medicine and Innovative Therapeutics, Murdoch University, Perth, WA 6150, Australia

**Keywords:** major histocompatibility complex (MHC), human leucocyte antigen (HLA), SINE-VNTR-Alu (SVA), expression quantitative trait loci (eQTL), transcription elements (TEs), clustered transcription, Parkinson’s disease (PD)

## Abstract

SINE-VNTR-Alu (SVA) retrotransposons can regulate expression quantitative trait loci (eQTL) of coding and noncoding genes including transposable elements (TEs) distributed throughout the human genome. Previously, we reported that expressed SVAs and human leucocyte antigen (HLA) class II genotypes on chromosome 6 were associated significantly with Parkinson’s disease (PD). Here, our aim was to follow-up our previous study and evaluate the SVA associations and their regulatory effects on the transcription of TEs within the HLA class II genomic region. We reanalyzed the transcriptome data of peripheral blood cells from the Parkinson’s Progression Markers Initiative (PPMI) for 1530 subjects for TE and gene RNAs with publicly available computing packages. Four structurally polymorphic SVAs regulate the transcription of 20 distinct clusters of 235 TE loci represented by LINES (37%), SINES (28%), LTR/ERVs (23%), and ancient transposon DNA elements (12%) that are located in close proximity to HLA genes. The transcribed TEs were mostly short length, with an average size of 389 nucleotides. The numbers, types and profiles of positive and negative regulation of TE transcription varied markedly between the four regulatory SVAs. The expressed SVA and TE RNAs in blood cells appear to be enhancer-like elements that are coordinated differentially in the regulation of HLA class II genes. Future work on the mechanisms underlying their regulation and potential impact is essential for elucidating their roles in normal cellular processes and disease pathogenesis.

## 1. Introduction

The coding capacity of the human genome is two to four percent for proteins and fifty percent or more for potentially transcribing transposable elements (TEs) and other noncoding RNA sequences, reflecting the evolutionary and ongoing impact of TEs on genome structure and function [1,2]. Two major classes of active TEs, the class I retrotransposons and the class II DNA transposons, may move or copy themselves to new positions within the genome. The class I TEs move via an RNA intermediate and include long terminal repeat (LTR)/endogenous retrovirus (ERV) retrotransposons, SINEs (short interspersed nuclear elements), LINEs (long interspersed nuclear elements), and SINE-VNTR-Alu (SVA) composite elements, whereas the class II TEs move directly as DNA elements through a ‘cut-and-paste’ mechanism [3]. While some autonomous transposition continues, most TEs in the human genome are immobile and have become highly mutated, fossilized (fixed) and/or fragmented over evolutionary time [4,5]. However, over eighty percent of the human genome might be transcribed with a significant portion of this transcriptional activity attributable to TE sequences [6]. The pervasive transcription of TEs continues to raise important considerations about their potential regulatory functions and genetic actions [7,8,9,10].

Recent advances in RNA sequencing technologies, comprehensive databases, and sophisticated computational analyses have revolutionized our understanding of TEs and their roles in gene regulation, genome stability, and evolution [11,12,13,14], providing insights into their diversity [15,16,17] and importance in health and disease [18]. For example, aberrant activation of TEs, such as Alu, L1, LTR, and SVA, can disrupt gene function, and genome stability and has been linked to cancer development [7,13,19], neurodegenerative diseases like amyotrophic lateral sclerosis (ALS) [20], psychiatric disorders [21], Alzheimer’s disease [22], and X-linked dystonia parkinsonism [23]. TEs influence immune responses, both by direct effects on immune-related genes [24,25] and by generating TE-derived antigens [26,27]. TE-derived RNAs might affect the gene expression of nearby genes, activating or repressing them through various mechanisms, including exonization [7,20], by serving as alternative promoters, enhancers, or insulators; contributing to spliceomics, epigenetic modifications, and chromatin structure alterations; or participating in RNA interference pathways [1,2]. Furthermore, structural retrotransposon insertion polymorphisms (RIPs) have been identified as expression quantitative trait loci (eQTL), that is, as presence or absence genotypes that can affect differential gene expression and influence the progression of disease [18,28,29]. The expressed functions of many millions of individual TE RNAs from a wide variety of families and subclasses [16], however, still remain largely unknown or have not been identified and investigated adequately [30].

Many structurally polymorphic SVAs that contribute to genetic variation have been associated with differential regulation of gene expression across the human genome including genes within the major histocompatibility complex (MHC) region located on chromosome 6 [31,32,33,34,35,36]. The MHC class II region, contains various duplicated HLA class II genes such as *HLA-DR, -DQ*, and *-DP*, which are highly polymorphic and essential for diverse antigen presentation by macrophages, dendritic cells, B cells and some endothelial and thymic cells to CD4 T cells [37,38]. This MHC region includes the transporter-associated with antigen processing 1 and 2 (*TAP1* and *TAP2*) genes and the proteosome subunit β types 8 and 9 (*PSMB8* and *PSMB9*) genes, which have a vital role in antigen processing as part of the adaptive immune response against various pathogens [39]. Regulation within this region is complex and involves multiple layers of control, including the influence of four structurally polymorphic SVA on the transcription levels of HLA classical and non-classical class II genes in the blood cells of a large number of subjects from the Parkinson’s Progression Markers Initiative (PPMI) cohort [35,36]. The two SVA insertions, NR_SVA_380 and R_SVA_27 near the *HLA-DRB1* gene, were inferred to modify the transcription of 22 genes including 9 to 11 in the class II region, as well as some other genes in the MHC class I and III regions. In comparison, R_SVA_85 and NR_SVA_381, inserted between the *HLA-DOA* and *-DPA1* genes, influenced only four genes in the MHC class II region. Some of these expressed SVAs and HLA class II gene alleles have been associated with Parkinson’s disease [35]. However, the most significant allelic differences between Parkinson’s disease (PD) and healthy cases after Bonferroni corrections were detected only for the expressed *HLA-DRA*01:01:01* and -*DQA1*03:01:01* alleles and the NR_SVA_381 genotype. The SVAs that regulated HLA gene transcriptional activities also were allele and haplotype dependent [35]. For example, of the 194 *DRB1*15:01/SVA_27* haplotypes in PPMI, 178 (91.8%) were linked to *DQA1*01:02/DQB1*06:02*.

Since TEs can influence immune responses [24,25,26,27], we hypothesized that the MHC class II SVA genotypes that regulate HLA class II genes might also co-regulate the transcription of particular TE families in the MHC class II region. What TE families and loci, if any, are co-expressed with HLA genes in the MHC class II region have not been investigated previously. We used the same PPMI cohort of 1530 individuals as previously reported [35] to reveal that four SVAs regulate the transcription of clusters of different TEs that are located on either side of the HLA class II genes. The TE RNAs were detected in the transcriptomes of peripheral blood cells using the software packages provided by Bioconductor for R [40]. Various other software packages including Matrix-eQTL [41] were used for false discovery rate (FDR)-corrected *p*-values and logistic regression β effects to assess SVA differential associations and regulatory effects on TE expression. The study results imply that clusters of TE transcripts expressed in association with HLA class II genes may have regulatory gene expression functions at the transcription, translation, and epigenetic levels.

## 2. Materials and Methods

RNA sequences within the blood transcriptome of a PPMI cohort were reanalyzed and TE sequences were identified and annotated from the RNA-seq datafiles that we previously prepared for 1530 individuals within the PPMI cohort of a published study of SVA regulation of HLA and non-HLA genes within the MHC genomic region without assessing any differences between the cases and controls [35,36]. The MHC class II genes in the RNA-seq data of 1530 individuals downloaded from the PPMI website (www.ppmi-info.org/, accessed 8 September 2024) were detected and counted by the arcasHLA software v0.6.0 [42]. The *GenomicAlignments* and *summarizeOverlaps* functions from the Bioconductor project [40] were applied in R to detect TE expression in the PPMI RNA-seq data using BAM RNA files in association with a TE-annotations file downloaded from the Hammell laboratory （https://labshare.cshl.edu/shares/mhammelllab/www-data/TEtranscripts/TE_GTF/, accessed 8 September 2024). The GTF input file of TE-annotations contained *class_id, family_id*, and a unique *transcript_id* (e.g., L1Md_Gf_dup1) that were assigned to each of the corresponding TE RNA sequences by *GenomicAlignments*. The annotated and counted TE RNA reads for each PPMI individual within the CSV output file was used for additional analyses.

Reference and non-reference SVAs and other TE genotype RIPs, and the regulatory effects of SVA on gene transcription levels were detected with MELT, Delly2, and DESeq2 software tools as described previously [18,33]. Transcript-based analysis of pair-ended Fastq files was performed using Salmon software v1.10.1 [43] and the outputs were reformatted for the Matrix-eQTL analysis [41], which calculated the statistically significant genetic loci of SVA regulating the expression transcript variants. Statistical outputs included FDR-corrected *p*-values and logistic regression β effects to assess SVA associations and regulatory effects on TE expression. The results that only remained significant after FDR correction are reported here (Appendix A).

Additional tables, counts, statistical averages, standard deviations, plots, graphs, and charts were performed using Excel and PowerPoint (Microsoft v16.78). The online program SRplot [44] was used for ridge line plots, PCA and scatter plots (https://www.bioinformatics.com.cn/srplot, accessed 8 September 2024).

## 3. Results

### 3.1. HLA Class II Gene Transcription Levels in Whole Blood Samples of 1530 Subjects of the PPMI Cohort

The transcription of nineteen MHC class II genes were detected and counted using arcasHLA software v0.6.0 [42] from within the bulk RNA sequence dataset of the PPMI cohort (Table 1). The transcription of the six classical (*HLA-DPA1, -DPB1, -DQA1, -DQB1, -DRA*, and *-DRB1*) and four non-classical (*HLA-DMA, -DMA, -DOA*, and *-DOB*) MHC class II genes were detected in all 1530 samples with an average read of 1864 counts per transcribed gene. The highest average read was for *HLA-DRA* (4421 counts) and the second highest was for *HLA-DRB1* (3772 counts). *DMA, DOA*, and *DOB* were low expression HLA genes with an average read between 143 and 604 counts per transcribed gene. The transcription of *HLA-DRB3*, *-DRB4*, and *-DRB5* genes that appear to be alleles of a single locus (DRB3,4,5) were represented by 1257, 929, and 600 samples, respectively, or 2786 samples when added together. This discrepancy between 1530 and 2786 samples suggests that the software counted transcripts from different haploids of the diploid DRB3,4,5 loci of the 1530 samples, whereas the transcript total counts from the other MHC class II loci are represented by diploid genes when the software could not differentiate between HLA allele differences at the gene loci. The average transcription level of the *HLA-DRB3*, *-DRB4*, and *-DRB5* genes ranged between 354 counts for *DRB3* and 1141 counts for *DRB5*. The lowest transcription reads of less than 15 counts per gene were for *HLA-DRB2*, *-DRB7*, *-DQA2*, *-DQB2*, *-DPA2*, and *-DPB2* (Table 1).

### 3.2. HLA Genes and TE Transcriptome Clusters Regulated by SVA

Figure 1 shows the organization and locations of 19 HLA class II genes, HLA pseudogenes, the *TAP* and *PSMB* genes, the *BRD2* gene, and the 20 clusters of transcribed TEs, C1 to C20, that are modulated by the four SVAs, NR_SVA_380, R_SVA_27, R_SVA_85, and NR_SVA_381, within the MHC class II genomic region. Appendix A lists the expressed MHC class I and class II genes and pseudogenes modulated statistically by the four SVAs. The table also provides the statistical outputs (statistic, *p*-value, FDR, and β-effect) by the Matrix-eQTL analysis and the chromosomal coordinated positions of the expressed MHC genes. Appendix A lists the 352 TEs within the twenty clusters C1 to C20 with statistical measures revealing their modulation by the four SVAs, NR_SVA_380, R_SVA_27, R_SVA_85, and NR_SVA_381. The expressed TE groups according to TE name, family, and class; positive or complementary DNA strand location; SVA effect (β) on individual TEs; and the number of different SVAs that regulate the same TE transcript are shown in Appendix A.

Table 2 summarizes the 20 TE-RNA clusters listed in Appendix A with the genomic coordinates of the 20 TE cluster positions, the TE cluster distance to its regulatory SVA element, and the DNA strand bias for transcribed TEs within and between clusters. There is considerable strand bias for transcribed TEs in at least 13 different clusters with 8 of 18 (44%) clusters at 100% of *cis* transcription and another five (28%) at 75% or greater. Also, more than 60% of TEs within particular clusters were expressed in the same orientation (*cis*) as the SVA integration. The presence of sense–antisense *Alu* in clusters C6, C7, C9, and C12, and sense–antisense L1 fragments in various clusters (Appendix A) are noteworthy because they have a potential to form complementary dsRNA with regulatory actions [45].

Table 3 provides the number and percentage of expressed TEs within clusters relative to the total number of TEs in the corresponding clusters of reference genome (GRCh38). Table 4 provides the cluster distance (bp) of the expressed TEs to the nearest gene, and the overall cluster loci and nucleotide length of each cluster locus. Table 5 lists the distance between the different adjoining transcribed TE clusters associated with the regulatory SVAs. Twelve of the twenty TE clusters occurred within the HLA-DRB/DQ haplotype region between *HLA-DRB9* and centromeric of the *DQB1* loci of 296.3 kb. Five of the TE clusters were between *HLA-DOB* and *BRD2* and mainly in the *TAP2* and *PSMB8* region. The other 3 of 20 clusters were in the *HLA-DPA1* and *-DPB1* gene region; altogether, 20 clusters of 235 TE loci span a distance of 652.7 kb across the MHC class II region.

Table 6 presents the number, percentage, length (bp), and classifications (class, family, and name) of the 235 transcribed TEs within the twenty clusters regulated by the four SVAs. At least 171 of the 352 TEs listed in Appendix A were regulated by two or more different SVAs so that only 235 loci were uniquely transcribed with overlapping regulation by one or other of the four SVAs. All the expressed TEs within the MHC class II region were represented by varying percentages of four TE classes, LINE (38%), SINE (28%), LTR (23%), and DNA (12%) (Figure 2). Of the 87 LINEs (L1 and L2), the L1 family was the most frequent at 77%. The 66 SINEs were 73% of *Alu* and 27% of MIR, with the 48 *Alu* divided into three subgroups, *AluJ* (8%), *AluS* (26%), and *AluY* (16%). The 53 LTRs were represented by four families, ERV1 (43%), ERVL-MaLR (36%), ERVK (15%), and ERVL (6%). Each of these families were divided into further RepeatMasker subgroup names, listed in Table 6 and Appendix A. The 28 DNA elements consisted of five families, with hAT-Charlie (29%), TcMar-Tigger (25%), and hAT (25%) being the three most frequent. The LINEs and SINEs were found in 18 of the 20 clusters, whereas the LTR and DNA elements were in 14 and 9 clusters, respectively. Surprisingly, almost all of the TEs were short-length, transcribed fragments with an average length of only 389 nucleotides. The HERVK3-int_dup98 that is located in cluster C10 telomeric of the *HLA-DRB1* gene and upregulated by R_SVA_27 was the longest fragment at 3210 bp. There were only six of 63 L1 fragments over a kilobase in size (between 1 kb and 3 kb); the majority were highly fragmented with an average size of 491 nucleotides. This average size is at least a tenth of the normal size of a full-length intact L1 sequence of 6–8 kb. Moreover, the smallest fragment in the TEs list was L1M4_dup6461 in C16, upregulated by NR_SVA_380 near the *HLA-Z* pseudogene centromeric of the *TAP1* and *PSMB9* genes. Figure 2 shows density plots for the length of the TE subfamilies *Alu*, MIR, DNA transposons, L1, L2, and LTR_ERVs. Table 7 lists the 16 of the 235 transcribed TE fragments that are longer than 1 kb in size (width). Eleven of them were upregulated and one was downregulated by R_SVA_27.

### 3.3. TE Expression and SVA Regulatory Effects within Four Subregions of the MHC Class II Genomic Region

The transcribed TEs were organized into 20 distinct clusters, C1 to C20, and four distinct subregions based on their genomic location and orientation, and proximity to HLA class II genes and/or non-HLA genes or pseudogenes (Table 2). Figure 3, Figure 4, Figure 5 and Figure 6 show four subregions with the locations of the 235 expressed TEs within clusters C1 to C20 together with the up- and downregulatory effects of the SVA elements using overlayed copies of different UCSC browser windows (https://genome.ucsc.edu/cgi-bin/hgGateway, accessed 8 September 2024).

Subregion A (C1 to C10) is the HLA-DRB region that is regulated by all four SVAs ranging across 129.4 kb from the *HLA-DRB9* pseudogene to the 5′ end of the *HLA-DRB1* classical class II gene (Figure 3). NR_SVA_380 is inserted within this region and modulates all ten clusters, upregulating all the TEs within clusters C1, C5, and C8, and downregulating all or most of the TEs in clusters C2, C4, C6, C7, and C9 and within particular sub-loci of cluster C10. R_SVA_27 modulates all clusters except for C1 and C3, upregulating all or most of the TEs within clusters C5, C6, C7, and C8 and within the latter half of C10 including nine upregulated TEs in the introns of *HLA-DRB1*. The longest cluster was C6 with 40 full-size or fragmented TEs upregulated by R_SVA_27, and 20 TEs downregulated by NR_SVA_380 at the 3′-end of *HLA-DRB5*. In contrast, R_SVA_85 and NR_SVA_381 regulated only a few of the TEs in clusters C4, C7, C9, and C10; R_SVA_85 downregulated all the TEs, whereas NR_SVA_381 upregulated the same TEs.

Subregion B (C10 to C12) is regulated by NR_SVA_380 and R_SVA_27 across 119.1 kb from the 5′ end of the *HLA-DRB6* pseudogene to the 5′ end of the *HLA-DQB1* gene, including HERVK3, *HLA-DRB1*, AluDRB1, *HLA-DQA1*, and *HLA-DQB1-AS1* (Figure 4). R_SVA_27 is inserted in this region at the 5′-end of the *HLA-DRB1* gene and upregulates most of the expressed TEs in these clusters including those in cluster C12, which was subdivided into C12a, C12b, and C12c because some of the TEs in C12a overlapped *HLA-DQA1*, C12b was intergenic between *HLA-DQA1* and *HLA-DQB1*, and C12c was intragenic or centromeric of *HLA-DQB1*. NR_SVA_380 regulated the expressed TEs only in C10, C12a, and C12b. Both SVAs upregulated the structurally polymorphic AluDRB1 that is located between C10 and C11, and that has been associated with *HLA-DRB1* alleles and used as a genetic marker in human population diversity studies [46,47]. In contrast, R_SVA_27 upregulated, while NR_SVA_380 downregulated the expression of AluYa5 that is located in intron 5 of *HLA-DRB1*.

Subregion C (C13 to C16) is regulated only by NR_SVA_380 across 98.7 kb from 5′ of *HLA-DQB2* to 5′ of *HLA-DMB* including *HLA-DOB* and the *TAP2* genes and a 20.1 kb area bordering the *HLA-Z* pseudogene (Figure 5). All of the TEs (except for L1MC5_dup5283 in C15) were downregulated and oriented mostly on the positive DNA strand.

Subregion D (C17 to C20) is regulated by NR_SVA_85 and R_SVA_381 across 135.2 kb from *HLA-DMA* to *HLA-DPA2* including an area 5′ of *HLA*-*DRB2* and the *HLA-DPA1* and *-DPB1* genes (Figure 6). Here, the same 27 TEs in four clusters were regulated by NR_SVA_85 and R_SVA_381, but in the opposite directions. So, whereas NR_SVA_85 downregulated the TEs in C17, C19, and C20, R_SVA_381 upregulated these TEs. On the other hand, NR_SVA_85 upregulated the four TEs in C17, whereas R_SVA_381 downregulated them. It is noteworthy that the TEs in C17 are immediately 5′prime of the *BRD2* gene coding for the bromodomain-containing protein 2 that is a transcription regulator and associates with transcription complexes and acetylated chromatin during mitosis.

Differential expression levels of TEs at different loci calculated by the Matrix eQTL software Version 2.3 provided two values, ‘statistic’ and ‘β’, for evaluating the SVA positive or negative regulation of TE transcription activity (Appendix A). Scatter plots of the effect size (β) versus statistical significance (‘statistic’) produced linear positive correlations for each of the SVA effects with highly varying squared correlation coefficients ranging between r^2^ of 0.11 for R_SVA_27 and 0.34 for NR_SVA_380, and 0.68 for R_SVA_85 and 0.7 for NR_SVA_381. Principle component analysis (PCA) of these two values in scatter plots showed that the ‘statistic’ value was the first principal component (PC1) that captured >80% of the total variance in the datasets, and the ‘β’ effect (PC2) captured the remainder of the variance at >15% (Figure 7). The ‘statistic’ combines both the ‘β’ value and its variance to provide a more comprehensive measure of the reliability and impact of the SVA effects on TE expression, show the main differences between the SVA effects, and to distinguish between the different groups or clusters in our data. In comparison, ‘β’, which represents the positive and/or negative effect size, captures less variance in the data, and therefore might not adequately separate distinct clusters. This emphasizes the importance of considering both effect size (‘β’) and significance (‘statistic’) when interpreting the regulatory impact of SVAs on TE expression. Thus, taken together, PC1 versus PC2 reveals that although the regulatory effects of the four SVAs on TE transcription are markedly different, there also is an overlap between the effects of particular pairs of SVAs.

Scatter plots of the ‘β’ effect size versus genomic position of the TE transcription activity also show that the differential effects of the four SVAs on TE transcription occurred within distinct groups or clusters (Figure 8 and Figure 9).

Overall, the four SVAs regulated the expression of TEs at 236 annotated loci. R_SVA_27 and NR_SVA_380 regulated the expression of 208 loci, and 73 of these loci were regulated by both SVAs. R_SVA_27 regulated 146 TE expressed loci, 119 upregulated (36 L1, 31 *Alu*, 19 ERV1, 3 ERVK, 4 hAT, 9 ERVL, 5 L2, 6 MIR, 5 TcMar, and 1 SVA), and 27 downregulated (9 L1, 4 L2, 4 *Alu*, 4 hATs, 2 MIR, 2 TcMar, 1 ERV1, and 1 ERVL). In comparison, NR_SVA-380 regulated 129 TEs, 44 upregulated (17 L1, 11 *Alu*, 4 ERV1, 3 ERVK, 3 hAT, 2 ERVL-MaLR, 2 TcMar-Charlie, 1MIR, and 1 L2) and 85 downregulated (20 L1, 16 *Alu*, 13 L2, 9 ERVL and ERVL-MaLR, 8 hAT, 6 MIR, 5 ERV1, 5 TcMar, and 2 ERVK). Thirteen of fourteen L2s were downregulated by NR-SVA_380, whereas only four of nine L2s were downregulated by R_SVA_27. The TEs upregulated by NR_SVA_380 and R_SVA_27 are shown as horizontal bar plots in Appendix A, respectively. Some of the TEs downregulated by NR_SVA_380 were upregulated by R-SVA_27 as seen in cluster C6 (Figure 3). Of the 73 co-expressed loci regulated by both R_SVA_27 and NR_SVA_380, 54 (74%) were downregulated by NR_SVA_380, whereas only 21 (29%) were downregulated by R_SVA_27. Figure 10a shows an XY scatter plot between the R_SVA_27 and NR_SVA_380 β effects for the 73 co-regulated TE loci. A significant (*p* < 0.01) positive linear relationship was obtained with R^2^ = 0.141.

R_SVA_85 regulated 39 TEs, 6 upregulated (2 *Alu*, 2 L1, 1 MIR, and 1 ERVL-MaLR) and 33 downregulated (8 L2, 6 MIR, 5 L1, 5 *Alu*, 4 hAT-Charlie, 3 ERVL-MaLR, 1 TcMar, and 1 ERVK). NR_SVA_381 regulated 38 TEs, many the same as R_SVA_85, but with opposite effects, 33 upregulated (8 L2, 6 MIR, 5 L1, 5 *Alu*, 4 hAT-Charlie, 3 ERVL-MaLR, 1 TcMar, and 1 ERVK) and 5 downregulated (1 *Alu*, 2 L1, 1 MIR, and 1 ERVL-MaLR). The TEs upregulated by NR_SVA_381 and R_SVA_85 are shown as horizontal bar plots in Appendix A, respectively. Figure 10b shows an XY scatter plot with a significant (*p* < 0.001, R^2^ = 0.999) negative relationship between the R_SVA_85 and NR_SVA_381 β effects for 30 co-regulated TE loci in the *HLA-DP* region.

### 3.4. Visualization of the Association between Transcribed TE Clusters and Candidate Cis-Regulatory Elements (cCREs) with the UCSC Online Browser

Figure 3, Figure 4, Figure 5 and Figure 6 show that the TE clusters are mostly intergenic, located between genes, pseudogenes, or lncRNA, although some intragenic (intronic) TEs are within at least eight of the HLA class II genes, *HLA-DRB9, -DRB5, -DRB6, -DRB1, -DQA1, -DOB, -DPA1*, and *-DPB1*. The H3K27Ac track in the figures indicates the chromatin influence on transcription regulatory regions and their close proximity to the SVA-regulated TE clusters. Overlays of the four subregions of transcribed TEs within the UCSC browser in relation to tracking locations and orientation of genes and the layered H3K27Ac histone enrichment from ENCODE reveal additional regulatory features and characteristics about the SVA regulation of TEs within the 20 clusters. Appendix A shows that the positions of expressed TE sites are highly concentrated and associated in close proximity to many epigenetic regulators including more than 132 enhancer and promoter sites, over 71 DNase I hypersensitivity peak cluster sites (Appendix A), and numerous H3K27Ac regions (UCSC genome browser, https://genome.ucsc.edu, accessed 8 September 2024). The transcribed TEs regulated by the SVAs are broad categories of different SINEs, LINEs, LTR/ERVs, and DNA transposons whose exact functions are not known, but some are likely to have enhancer and suppressor like functions, which generate eRNA (enhancer-transcribed/derived RNA).

## 4. Discussion

RNA sequencing of blood cells from more than 1500 individuals in the PPMI cohort revealed regulatory effects of hundreds of SVA loci on TE RNA transcription across the entire genome [18,33,34,36,48]. We took a small subset of this larger dataset and examined the effect of four structurally polymorphic SVAs on 20 TE RNA clusters spanning the genomic region of 15 HLA class II genes from the telomeric pseudogene *HLA-DRB9* to the centromeric pseudogene *HLA-DPB2* in 0.63 Mb of the HLA class II region. This HLA genomic region is one of the most gene-dense regions of the human genome that is associated with numerous human diseases and includes the antigen processing genes, *TAP1* and *TAP2*, and *PSMB8* and *PSMB9*, and the transcriptional regulator gene, *BRD2*, that encodes for the BET (bromodomains and extra terminal domain) family of proteins [38,49]. Overall, the MHC class II genomic region with transcription activity at 235 TE loci in 20 delineated clusters regulated by four SVA elements is ~0.021% of the entire human genome (~3.2 × 10^9^). In comparison to the MHC class II genomic region, detectable TE RNA cluster activity was absent within 0.88 Mb of the entire MHC class III region with only a single L2 (281 bp) element upregulated by NR_SVA_380 located within the *TSBP1-AS1* sequence.

The insertion frequencies of NR_SVA_380, R_SVA_27, R_SVA_85, and NR_SVA_381 were reported as 0.26, 0.24, 0.98, and 0.36, respectively, in the individuals of the PPMI cohort [35,36]. The number of individuals that might carry one or more of the low frequency SVA insertions within their diploid cells (individuals with two haplotype genomes) is not known, but the high frequency R_SVA_85 is almost completely fixed within the human genome and is likely to be found in more than 98% of individuals. In the present study, we found that NR_SVA_380 regulated the expression of 16 TE clusters and 129 TE loci, compared to 10 clusters and 148 TE loci by R_SVA_27, 8 clusters and 37 TE loci by R_SVA_85, and 8 clusters and 38 TE loci by NR_SVA_381 within the MHC class II genomic region. Some of the same expressed TE loci were regulated by two or more different SVAs. For example, 83 of 205 expressed TEs were either up- or downregulated by both NR_SVA_380 and R_SVA_27 (Figure 8). On the other hand, 30 of 31 TEs were regulated by both R_SVA_85 and NR_SVA_381, but mostly downregulated by SVA-85 and upregulated by SVA_381 (Figure 9). This result reveals the highly coordinated regulation of the expressed TEs by SVAs in the MHC class II region. Also, not all of the TE loci within the TE cluster regions are expressed (ranging between 6 and 100%, average of 54%) when compared to the TEs in the reference genome, and, therefore, the SVA regulatory effect on TE transcription is selective (Table 2 and Figure 3, Figure 4, Figure 5 and Figure 6).

Most TE clusters overlap HLA genes or begin or end within 5kb of the genes. The exceptions are C4, C5, C8, and C11, which range from 6.1 kb to 12.7 kb from the gene core (Table 3). Based on their locality, these expressed TE clusters probably have a role in regulating the gene expression, since there are many candidate regulatory enhancers (CREs) within these gene regions and also beyond them. When interrogating the nascent transcription of the functionally related genes clustered within the mouse MHC genomic region, Mahat et al. [50] found that multiple enhancers correlated with each MHC gene. In our study, the expressed human TE clusters C1 to C3, C6 and C7, C9 and C10, and C10 overlapped regions of the *HLA-DRB9, -DRB5, -DRB6*, and *-DRB1* genes, respectively, in the DRB gene region (Figure 3). Clusters C11 and C12 incorporating up to 49 expressed TE loci are in close proximity and/or overlap the *HLA-DQ1* and -*DQB1* classical class II genes that encode molecules involved with antigen presentation to CD4 T cells. The five expressed TE clusters, C13 to C17, are noteworthy because they are in close proximity to at least three distinct recombination hotspots involved with HLA haplotype shuffling [37,51,52], and the clusters C13 to C16 are regulated by only NR_SVA_380. In this regard, the clusters C13 and C14 overlap parts of the *HLA-DOB* gene, whereas C15 includes the 3′-end of the *TAP2* gene. The DOB accessory protein encoded by the *HLA-DOB* gene does not present antigens extracellularly to T cell receptors, but instead binds with the DOA protein to suppress peptide loading of MHC class II molecules by inhibiting the HLA-DM accessory protein that is involved in intracellular antigen processing and presentation [53]. The five TE loci in C15 are slightly telomeric of a major recombination hotspot that was identified in intron 2 of the *TAP2* gene [54]. The six transcribed TEs within C16 are located between *PSMB9* and *HLA-DMB* and overlay the *HLA-Z* (88 bp) pseudogene and an uncharacterized lncRNA (~9590 bp) at locus *LOC100294145*. This TE cluster includes an extended transcribed Charlie1 element (1090 bp) with a 210 bp sequence of DNase hypersensitivity item 50 that is located 4 kb centromeric of *LOC100294145* (see Appendix A, and UCSC browser at https://genome.ucsc.edu, accessed 8 September 2024).

The 15 or 16 expressed TEs of C17, which are downregulated by R_SVA_85 and upregulated by NR_SVA_381, are located approximately midway between *HLA-DMB* and *BRD2* (Figure 1) and are flanked by two regions that are highly concentrated with many epigenetic regulators including more than 50 enhancer and promoter sites, over 28 DNase I hypersensitivity peak cluster sites, and numerous H3K27Ac regions (Appendix A). The HLA-DMA and -DMB heterodimer molecules are located in intracellular vesicles and play a central role with the attachment of peptides to MHC class II molecules by helping to release the CLIP molecule from the peptide binding site. The BET protein expressed by *BRD2* appears to be involved in many physiological and pathological processes including the immune response, has a role in nucleosome assembly and chromatin remodeling [55,56], and interacts genome-wide with the architectural/insulator protein CCCTC-binding factor (CTCF) to form transcriptional boundaries [57]. BET has been associated with transcription complexes and with acetylated chromatin during mitosis, and selectively binds to the acetylated lysine-12 residue of histone H4 via its two bromodomains [58]. *BRD2* expresses multiple alternatively spliced variants, and has been implicated in juvenile myoclonic epilepsy [59] and inflammatory bowel disease [60].

The transcribed TEs regulated by the SVAs belong to broad categories of different SINEs, LINEs, LTR/ERVs, and DNA transposons whose exact functions are not known, but are likely to have enhancer and suppressor like actions, which generate eRNA (enhancer-transcribed/derived RNA). Most of the twenty clusters of expressed TEs are short sequences or fragments of about 389 bp or less in regions of gene expression regulatory elements such as CREs (enhP, enhD, K4me3, prom, and CTCF), DNase hypersensitivity clusters, and H3K27Ac marks that are involved with nucleosome assembly and chromatin architecture that have been detected in ENCODE studies and surveys [61,62,63]. The many hundreds of ENCODE enhancers within the MHC class II region (Appendix A) displayed online using the UCSC browser are small (~300 bp in length), ranging on average from 200 to 1000 bp. These sequences can be located far from the gene that they regulate, either upstream, downstream, or within introns. At least 28% of the expressed TEs in our study were full-length SINEs (*Alu* and MIR) of ~300 bp, and 23% were solitary LTR sequences mainly < 520 bp (Figure 2). Although full-length LINE sequences are usually ~6 kb, most of the expressed L1 sequences (29% of 235 TEs) in this study were 3′-fragments with a peak density size of ~491 bp, a 3′ L1 fragment size that is known to bind to and regulate histone functions [64]. Also, the TE RNA clusters are located either within or between HLA class II gene regions or candidate regulator elements within the MHC class II genomic region as displayed with the UCSC browser and the ENCODE cCRE and ENCODE regulation tracks (Appendix A). Therefore, based on size, location, and transcription, we hypothesize that the clustered TEs are expression RNA enhancers (e-RNA), particularly as all classes of TE sequences have enhancer ability [29,65,66], and that enhancers are often transcribed as part of their regulatory mechanisms [67,68]. In addition, chromatin and DNA methylation might be master regulators and coordinators of TE expression enhancers and suppressors [1,7,69] that we observe in this study. The MHC methylation profiles have a bimodal distribution whereby the vast majority of the analysed regions were either hypo- or hypermethylated when correlated with independent gene expression data [70,71]. Future studies incorporating precise distance measurements between expressed TEs, gene transcription start sites and methylation profiles might help to differentiate between their particular roles as enhancers and super-enhancers in the coordinated regulation of the many duplicated HLA genes in the MHC class II region.

Since insertion polymorphic SVAs are associated with particular HLA haplotypes including TEs [37], future work might examine whether these TE RNA clusters are differentially associated with different HLA haplotypes. Three of the four SVAs in this study have population frequencies of less than 20% that are in strong linkage disequilibrium (LD) with particular HLA alleles and HLA-DRB/DQ/DP haplotypes. Some of the expressed TEs in this study are in the same genomic locations as fifteen structurally polymorphic TEs or indels such as AluDR1, AluDQA1(a), AluDQA1(B), AluLTR12.DRB5, AluMER66, MER11-DQB1, LTR14-DRB1, LTR42-DOB, and LTR-DOB (Appendix A) that were associated previously with particular HLA class II haplotypes in a panel of 95 homozygous EBV-transformed human B cell lines [35]. Two of these polymorphic *Alu* insertions, AluDRB1 and AluDQA1(a), were in LD with particular HLA alleles in different world populations [46,72] including with *HLA-DRB1* alleles in 12 minority ethnic populations in China [47]. Therefore, different SVA RIPs probably affect different TE insertion/deletion genotypes (Appendix A) and HLA alleles [35]. For example, R_SVA_27 (alias SVA-DRB1 in [37]), with a population frequency of 11.9%, is associated strongly (>90%) with genotypes (alleles) AluDRB1, AluDQA1.a, AluLTR12.DRB5, *HLA-DRB1*15/16*, SVA-DRB5, LTR14-DRB5, *HLA-DRB5*, and LTR5-DQB1. In contrast, NR_SVA_380 with a population frequency of 13.1% is associated with various other genotypes (alleles), AluMER66, *HLA-DRB1*01/10*, SVA-DRB1, *HLA-DRBnull3/4/5*, but shares the same AluDRB1(AluY_dup35538) insertion genotype with R_SVA_27 as an upregulated duplicate (Appendix A, [35,37]). In this study, we did not undertake a detailed analysis of the associations between the expressed TEs and HLA alleles or haplotypes, which is an added level of complication that requires a separate investigation. Future studies of the association between the expressed TEs and HLA haplotypes might provide additional insights into the roles and importance of these expressed candidate TE enhancers and suppressors.

Counting and annotating TE transcripts in RNA sequencing (RNA-seq) analyses presents several challenges to overcome such as sequence similarity, multiple mapping reads, low abundance of TE transcripts, and reference bias towards coding regions, biological variation, and library preparation artefacts. We addressed these challenges by employing specialized tools and pipelines designed for TE analysis, applying rigorous multi-mapping read handling techniques, and enhancing TE annotations in reference genomes [18,40,41,73]. We also used a TE genomic annotation file developed for the *TEtranscripts* software package [74] to assign a unique TE ID number (e.g., AluSx3_dup10815, AluSx1_dup35983, in Appendix A) to our TE annotated RNA sequences for cross-referencing between individual TEs in this and other RNA sequencing studies. Ultimately, confirmation and interpretation of these results will depend on additional comparative analyses and in vitro experimentation using cell lines.

This is the first report to have focused exclusively on expressed TEs in the MHC class II region. On the basis of the vast number and diversity of TEs and a lack of experimental data, it is not possible to conclude exactly what role if any that each of the particular TEs may have in the regulation of HLA gene expression. Many intracellular and extracellular factors such as infectious agents, chemicals, cytokines, hormones, and methylating agents might affect the coordinated expression of the TEs in association with the non-HLA and HLA class II genes [49,75,76]. Based on the results and insights of other studies in plants and animals, including human cells in vitro and in vivo, it is reasonable to assume that the transcribed TEs identified in our study have some role in the regulation of gene expression [7,65,77,78,79,80]. Although much of our understanding of transcriptional activity of transposable elements (TEs) within the MHC class II region is speculative at this time, this area of research provides opportunities for uncovering novel mechanisms of gene regulation and immune function, with potential implications for understanding and treating various diseases. Further experimental studies are essential to validate these speculative insights and translate them into practical applications.

## Figures and Tables

**Figure 1 genes-15-01185-f001:**
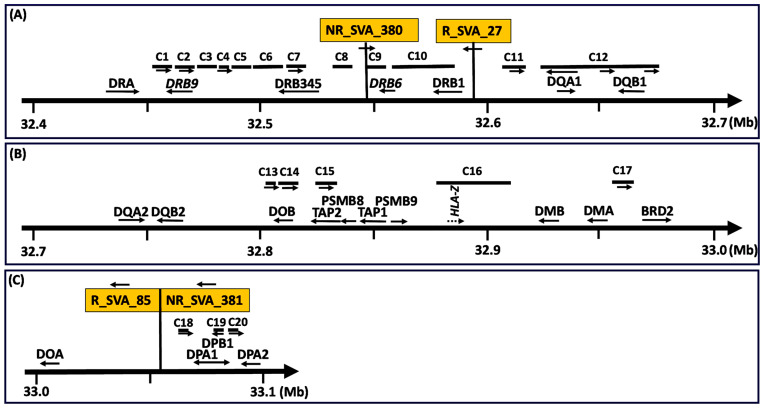
Map of the relative genomic positions of four regulatory SVA repeat elements, clusters of expressed TEs labelled C1 to C20, HLA class II genes, pseudogenes, and non-HLA genes (**A**–**C**). Telomeric to centromeric orientation of the genomic regions on chromosome 6 (chr6) is left to right, respectively. Distances (Mb) starting at 32.4 Mb from the telomeric end (panel (**A**)) and ending at 33.1 Mb (panel (**C**)) towards the centromeric end are indicated by the numbers beneath the horizontal thick arrows. C1 to C20 are locations of clusters of expressed TEs modulated by the SVA in orange boxes labelled as NR_SVA_380 and R_SVA_27 in panel (**A**), and R_SVA_85 and NR_SVA_381 in panel (**C**). The horizontal arrows below the SVA labelled boxes indicate the direction of the SVA sequence that is on the forward or reverse DNA strand. The arrows below the genes and some clusters indicate the orientation of the sequences either in the forward (towards the centromere) or reverse (towards the telomere) direction. Clusters without horizontal arrows are TE directions with mixed orientations.

**Figure 2 genes-15-01185-f002:**
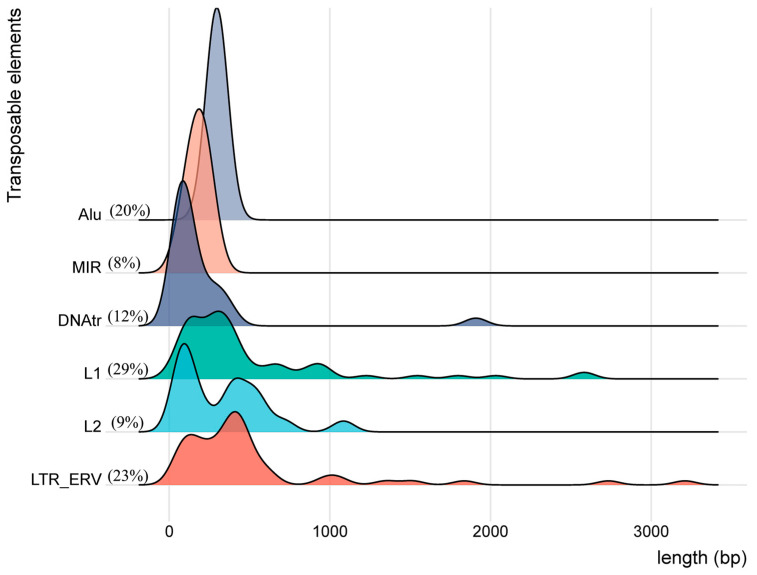
Density plot of the distribution of the length (bp) of TEs within each subfamily group, Alu, MIR, DNAtr, L1, L2, and LTR_ERV along the *X*-axis relative to their density on the *Y*-axis.

**Figure 3 genes-15-01185-f003:**
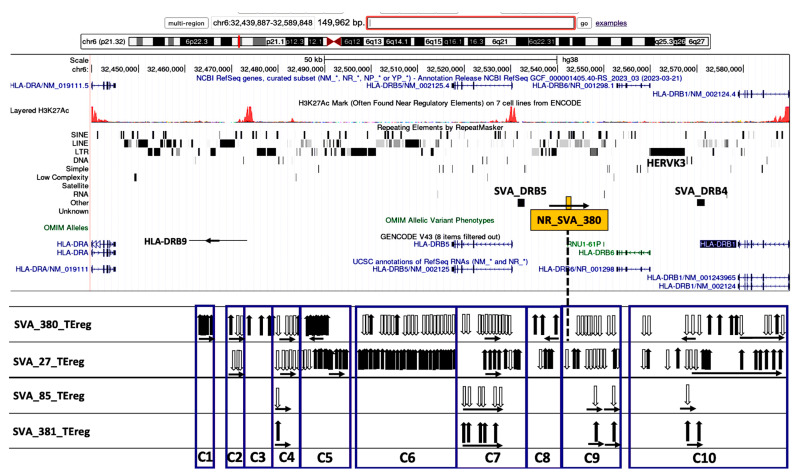
Genomic loci map of expressed TE clusters C1 to C10 ranging across 150 kb from the HLA-DRA to the 5′ end of the HLA-DRB1 classical class II gene that are regulated by four different SVAs. Image of the genome browser is sourced from the University of California, Santa Cruz (UCSC) Genomics Institute, showing from the top towards the bottom, the scale for chr6:32,439,887–32,589,846 and selected tracks for NCBI reference genes, H3K27Ac mark, repeating elements, Genecode gene annotations, and UCSC RefSeq RNAs. The browser image is overlayed with the positions of NR_SVA_380 (orange box), SVA_DRB4, and SVA_DRB5. Below the browser image are the relative positions of expressed TEs (vertical arrows) within the boxed clusters C1 to C10 that are regulated by the labelled SVA elements, SVA_380, SVA_27, SVA_85, and SVA_381, within each horizontal panel. The black vertical arrows indicate upregulated TEs, and white vertical arrows indicate downregulated TEs. The horizontal arrows in each cluster group below the vertical arrows indicate the forward (left to right) or reverse (right to left) orientation of the TE loci.

**Figure 4 genes-15-01185-f004:**
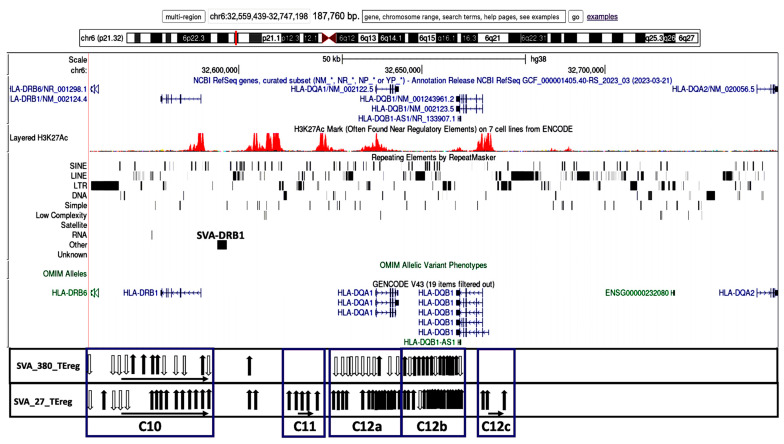
Genomic loci map of expressed TE clusters C10 to C12c within 187.8 kb from the HLA-DRB6 to the 5′ end of the HLA-DQA2 that are regulated by two SVAs. Image of the genome browser is sourced from UCSC Genomics Institute, showing from the top towards the bottom, the scale for chr6:32,559,439–32,747,198 and selected tracks for NCBI reference genes, H3K27Ac mark, repeating elements, and Genecode gene annotations. Below the browser image are the relative positions of expressed TEs (vertical arrows) within the boxed clusters C10 to C12c that are regulated by the labelled SVA elements, SVA_380, and SVA_27 within each horizontal panel. The black vertical arrows indicate upregulated TEs, and white vertical arrows indicate downregulated TEs. The horizontal arrows in each cluster group that are below the vertical arrows indicate the forward (left to right) or reverse (right to left) orientation of the TE loci.

**Figure 5 genes-15-01185-f005:**
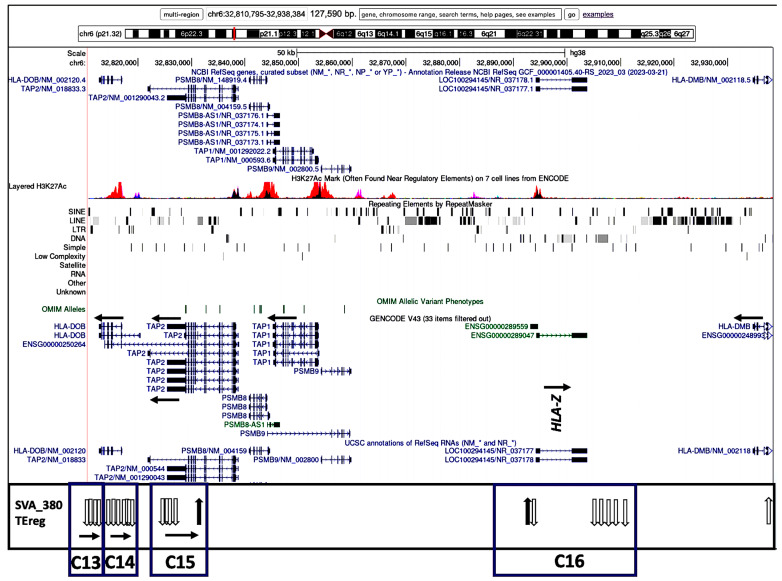
Genomic loci map of expressed TE clusters C13 to C15 within 127.6 kb from HLA-DOB, TAP2 to HLA-DMB and regulated by NR_SVA_380. Image of the genome browser is sourced from UCSC Genomics Institute, showing from the top towards the bottom, the scale for chr6:32,810,795–32,938,384, and selected tracks for NCBI reference genes, H3K27Ac mark, repeating elements, and Genecode gene annotations. The position of the HLA-Z pseudogene fragment is indicated. Below the browser image are the relative positions of expressed TEs (vertical arrows) within the boxed clusters C13 to C15 that are regulated by SVA_380. The black vertical arrows indicate upregulated TEs, and white vertical arrows indicate downregulated TEs. The horizontal arrows in each cluster group that are below the vertical arrows indicate the forward (left to right) or reverse (right to left) orientation of the TE loci.

**Figure 6 genes-15-01185-f006:**
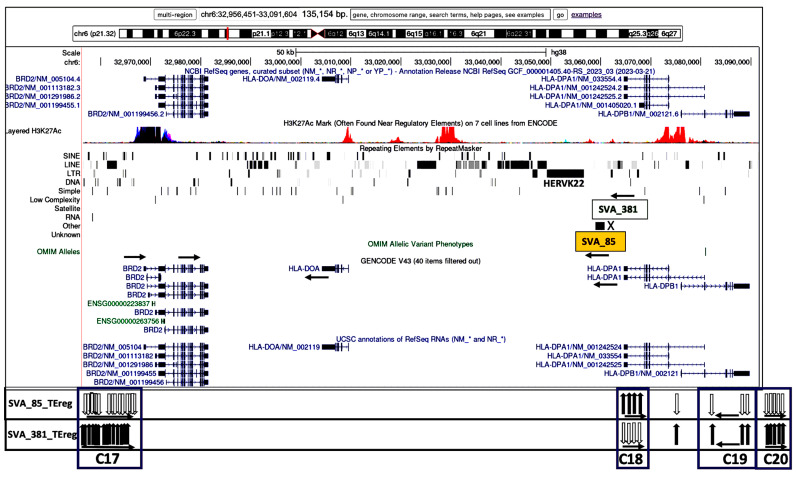
Genomic loci map of expressed TE clusters C17 to C20 within 135.2 kb from BRD2 to HLA-DPB1 and regulated by R_SVA_85 and NR_SVA_381. Image of the genome browser is sourced from UCSC Genomics Institute, showing from the top towards the bottom, the scale for chr6:32,956,451–32,091,604, and selected tracks for NCBI reference genes, H3K27Ac mark, repeating elements, and Genecode gene annotations. The locations of the R_SVA_85 (orange box) and NR_SVA_381 (open box) elements are indicated. Below the browser image are the relative positions of expressed TEs (vertical arrows) within the boxed clusters C17 to C20 that are regulated by SVA_85 and SVA_380 are indicated in each horizontal panel, respectively. The black vertical arrows indicate upregulated TEs, and white vertical arrows indicate downregulated TEs. The horizontal arrows in each cluster group that are below the vertical arrows indicate the forward (left to right) or reverse (right to left) orientation of the TE loci.

**Figure 7 genes-15-01185-f007:**
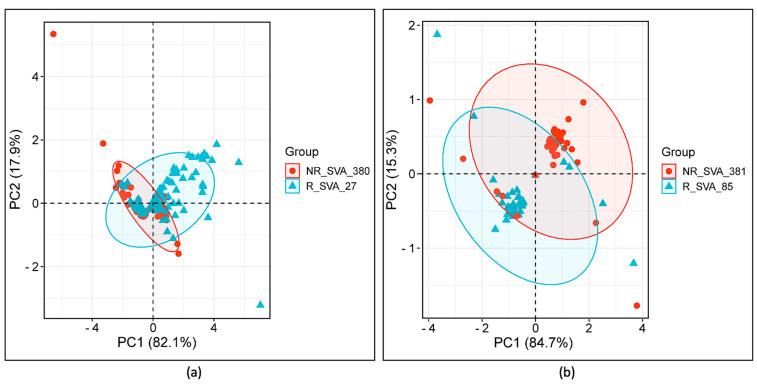
PCA plots of SVA comparative effects on TE expression using two eQTL values, statistic (PC1) and β (PC2). (**a**) Comparison of variance between the regulatory effects of NR_SVA_380 and R_SVA_27 for 146 TE transcript samples. (**b**) Comparison of variance between the regulatory effects of NR_SVA_381 and R_SVA_85 for 38 TE transcripts. Concentration ellipses highlight the 95% confidence intervals around the core clusters of TE data points. Samples not regulated by either SVA are located at (0,0).

**Figure 8 genes-15-01185-f008:**
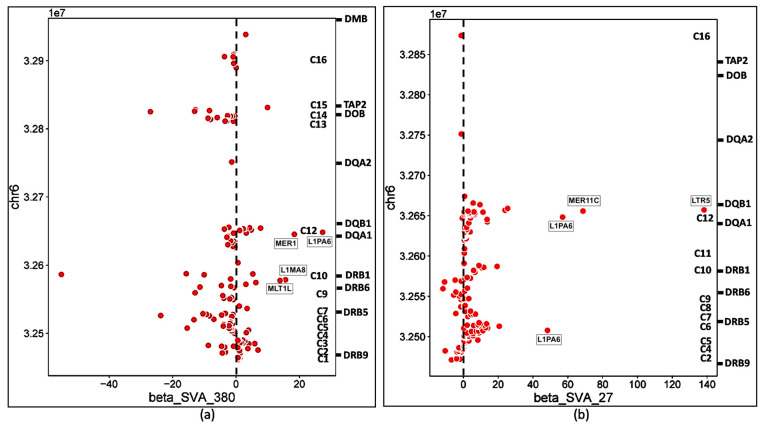
Scatter plots of NR_SVA_380 and R_SVA_27 effects on TE expression using the TE chromosomal position (*Y*-axis, units of 1x10^7^) compared to the β expression effect (*X*-axis). The regulatory effects (β) of NR_SVA_380 (**a**) and R_SVA_27 (**b**) are for 129 and 146 TE transcript samples, respectively. The relative gene locations and clusters C1 to C16 are indicated on the right-sided Y axis of (**a**,**b**). Some upregulated TE outlier red circles such as MER1, MER11c, LTR5, and L1PA6 are labelled within the (**a**,**b**) scatter plot matrices.

**Figure 9 genes-15-01185-f009:**
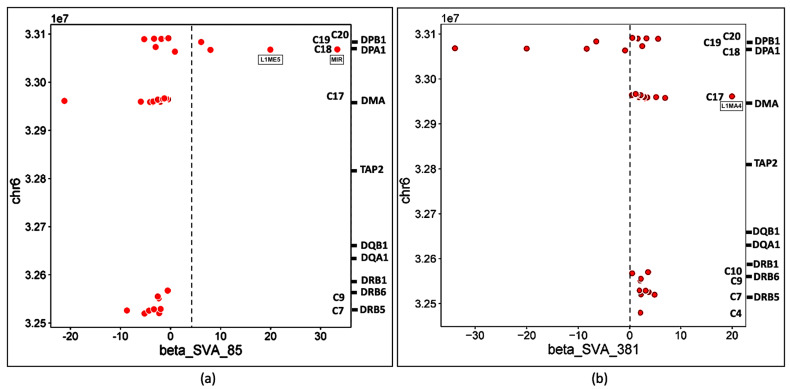
Scatter plots of (**a**) R_SVA_85 and (**b**) NR_SVA_381 effects on TE expression using the TE chromosomal position (*Y*-axis, units of 1x10^7^) compared to the β expression effect (*X*-axis). The regulatory effects (β) of R_SVA_85 (**a**) and NR_SVA_381 (**b**) are for 39 and 38 TE transcript samples, respectively. The relative gene locations and clusters C are indicated on the right-sided Y axis of each (**a**,**b**). Upregulated TE outlier red circles L1ME5 and MIR (**a**) and L1MA4 (**b**) are labelled within the scatter plot matrices, respectively.

**Figure 10 genes-15-01185-f010:**
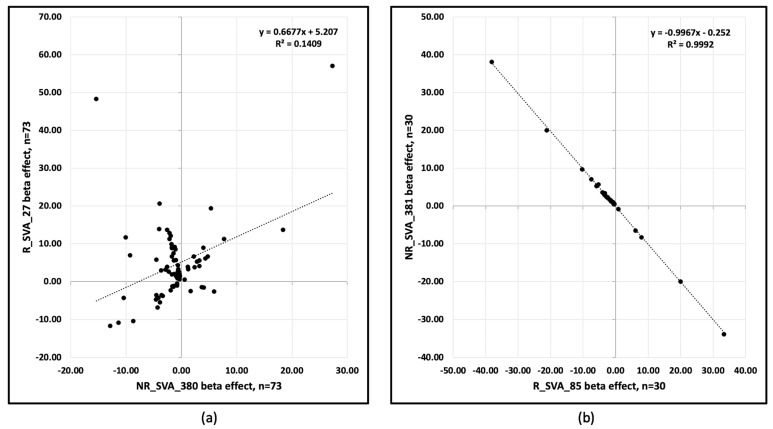
XY scatter plots of (**a**) R_SVA_27 β effect (*Y* axis) versus the NR_SVA_380 β effect (*X* axis) for 73 co-regulated TE loci, and (**b**) NR_SVA_381 β effect (*Y* axis) versus R_SVA_85 β effect (*X* axis) for 30 co-regulated TE loci. (**a**) R^2^ = 0.1409, *p* < 0.01; (**b**) R^2^ = 0.9992, *p* < 0.001.

**Table 1 genes-15-01185-t001:** HLA class II gene transcription (read counts) in blood cell RNA sequences of PPMI cohort.

HLA Gene	Mean RNA Count	Max Count	Min Count	STDEV * Count	Gene DNA Strand	Number Samples	% Total (1530) Samples
DRA	4420.8	20,991	150	2062.5	+	1530	100
DRB1	3771.5	17,018	295	1904.2	−	1530	100
DRB2	3.0	26	1	3.6	−	481	31
DRB3	353.5	2093	1	301.2	−	1257	82
DRB4	378.2	2043	1	281.7	−	929	61
DRB5	1140.9	7613	1	1144.3	−	600	39
DRB7	2.7	34	1	3.1	−	547	36
DQA1	1364.3	6479	50	840.0	+	1530	100
DQB1	1522.9	8785	83	1123.3	−	1530	100
DQA2	5.2	43	1	6.7	+	661	43
DQB2	7.7	92	1	11.7	−	793	52
DOB	277.6	2644	15	176.1	−	1530	100
DMB	1159.1	3740	50	467.4	−	1530	100
DMA	604.0	1990	28	242.6	−	1530	100
DOA	143.2	927	10	74.8	−	1530	100
DPA1	2621.7	7519	99	1186.7	−	1530	100
DPB1	2760.8	9181	141	1213.7	+	1530	100
DPA2	5.7	59	1	6.1	−	1417	93
DPB2	13.3	175	1	10.3	+	1523	99.5

* STDEV is standard deviation of the mean for cohort population samples.

**Table 2 genes-15-01185-t002:** Cluster genomic location, distance to regulatory SVA, and the DNA strand bias for transcribed TEs within and between clusters.

Regulatory SVA	Cluster	Cluster Genomic Location chr6:		Distance C to SVA	Number of Expressed TE Loci and DNA Positive or Negative Strand Bias			
		Start	End	bp	Number (+)	Number (−)	No. Pos and Neg Strands	% of Pos Strand
NR_SVA_380	C1	32,461,758	32,464,510	−82,325	5	0	5	100
chr6:32546835	C2	32,470,849	32,472,587	−74,248	3	0	3	100
	C3	32,474,505	32,479,598	−67,237	0	4	4	0
	C4	32,479,723	32,482,919	−63,916	4	0	4	100
	C5	32,484,596	32,490,660	−56,175	5	4	9	56
	C6	32,499,386	32,517,317	−29,518	9	11	20	45
	C7	32,519,925	32,529,312	−17,523	12	0	12	100
	C8	32,536,462	32,540,133	−6702	1	2	3	33
	C9	32,548,849	32,552,637	2014	2	3	5	40
	C10	32,555,239	32,587,951	8404	10	5	15	67
	C12a	32,626,240	32,642,457	79,405	3	9	12	25
	C12b	32,645,261	32,655,746	98,426	9	5	14	64
	C13	32,810,795	32,812,556	263,960	4	1	5	80
	C14	32,815,430	32,819,280	268,595	7	0	7	100
	C15	32,825,036	32,832,212	278,201	5	0	5	100
	C16	32,889,395	32,909,479	342,560	3	3	6	50
R_SVA_27	C2	32,471,170	32,472,587	−121,607	3	0	3	100
chr6:32594194	C4	32,479,723	32,483,752	−110,442	5	1	6	83
32596780	C5	32,484,596	32,492,552	−101,642	8	3	11	73
	C6	32,493,446	32,517,317	−76,877	20	20	40	50
	C7	32,524,854	32,532,999	−61,195	7	1	8	88
	C8	32,536,958	32,538,760	−55,434	1	2	3	33
	C9	32,546,087	32,552,637	−41,557	4	3	7	57
	C10	32,555,239	32,591,167	−3027	15	2	17	88
	C11	32,619,714	32,624,756	22,934	4	1	5	80
	C12a	32,625,903	32,635,931	29,123	5	10	15	33
	C12b	32,641,003	32,659,013	44,223	18	9	27	67
	C12c	32,663,614	32,674,360	66,834	3	0	3	100
R_SVA_85	C7	32,519,925	32,529,312	−529,634	7	0	7	100
chr6:33058946	C9	32,551,513	32,555,433	−503,513	2	0	2	100
33060797	C10	32,567,470	32,567,780	−491,166	1	0	1	100
	C17	32,956,451	32,966,714	−92,232	16	0	16	100
	C18	33,063,447	33,068,508	2650	3	1	4	75
	C19	33,083,458	33,089,755	22,661	0	3	3	0
	C20	33,089,799	33,091,604	29,002	5	0	5	100
NR_SVA_381	C4	32,479,723	32,480,033	−582,500	1	0	1	100
chr6:33062533	C7	32,519,925	32,529,312	−533,221	5	0	5	100
	C9	32,551,513	32,551,780	−510,753	1	0	1	100
	C10	32,555,239	32,571,178	−491,355	3	0	3	100
	C17	32,956,451	32,966,714	−95,819	15	0	15	100
	C18	33,063,447	33,068,508	914	3	1	4	75
	C19	33,083,458	33,089,755	20,925	0	3	3	0
	C20	33,089,799	33,091,604	27,266	5	0	5	100
				total	242 (69%)	107 (31%)	349 (100%)	

**Table 3 genes-15-01185-t003:** The number and percentage of expressed TEs within clusters relative to the total number of TEs in the clusters of reference genome (GRCh38).

Regulatory SVA	Cluster	Number Expressed TEs	Total TE on Ref Pos Strand	Total TE on Ref Neg Strand	No. TE in Reference	% of Ref Genome TE Expressed
NR_SVA_380	C1	5	5	1	6	83
	C2	3	4	0	4	75
	C3	4	1	5	6	67
	C4	4	5	1	6	67
	C5	9	11	4	15	60
	C6	20	18	18	36	56
	C7	12	16	3	19	63
	C8	3	5	4	9	33
	C9	5	5	7	12	42
	C10	15	28	19	47	32
	C12a	12	16	13	29	41
	C12b	14	23	5	28	50
	C13	5	4	1	5	100
	C14	7	7	2	9	78
	C15	5	8	2	10	50
	C16	6	13	23	36	17
R_SVA_27	C2	3	3	0	3	100
	C4	6	5	3	8	75
	C5	11	13	6	19	58
	C6	40	20	22	42	95
	C7	8	13	3	16	50
	C8	3	2	3	5	60
	C9	7	5	9	14	50
	C10	17	23	18	41	41
	C11	5	4	3	7	71
	C12a	15	10	11	21	71
	C12b	27	25	9	34	79
	C12c	3	11	4	15	20
R_SVA_85	C7	7	14	3	17	41
	C9	2	2	2	4	50
	C10	1	1	0	1	100
	C17	16	18	10	28	57
	C18	4	4	4	8	50
	C19	3	2	3	5	60
	C20	5	5	0	5	100
NR_SVA_381	C4	1	1	0	1	100
	C7	5	12	4	16	31
	C9	1	1	0	1	100
	C10	3	7	6	13	23
	C17	15	18	10	28	54
	C18	4	4	4	8	50
	C19	3	2	3	5	60
	C20	5	5	0	5	100
		349 (100%)	399 (62%)	248 (38%)	647 (100%)	54%

**Table 4 genes-15-01185-t004:** Cluster distance (bp) to nearest gene.

Cluster Number	Cluster Distance to Nearest Gene			Cluster Locus on Chromosome 6		
	Telomeric * End (bp)	Gene Locus Symbol	Centromeric * End (bp)	chr6 Start	chr6 End	C Length bp
C1	1937	DRB9	8990	32,461,758	32,464,510	2752
C2	−1005	DRB9	−913	32,470,849	32,472,587	1738
C3		DRB9	1005–6098	32,474,505	32,479,598	5093
C4		DRB9	6223–10,252	32,479,723	32,483,752	4029
C5		DRB9	11,096–17,160	32,484,596	32,492,552	7956
C6	17,967–36	DRB5		32,493,446	32,517,317	23,871
C7	−2572	DRB5	975	32,519,925	32,532,999	13,074
C8		DRB5	6175–9846	32,536,462	32,540,133	3671
C9	6626–0	DRB6		32,546,087	32,552,637	6550
C10	−2526	DRB6	27,949	32,555,239	32,591,167	35,928
C10	23,536	DRB1	−9176	32,555,239	32,591,167	35,928
C11	17,692–12,650	DQA1		32,619,714	32,624,756	5042
C12a	11,166	DQA1	−1227	32,625,903	32,642,457	16,554
C12b	−1577	DQA1	12,062	32,641,003	32,659,013	18,010
C12b	14,206–372	DQB1		32,641,003	32,659,013	18,010
C12c	−3043	DQB1	7703	32,663,614	32,674,360	10,746
C13	1968	DOB	−207	32,810,795	32,812,556	1761
C14	−2667	DOB	2278	32,815,430	32,819,280	3850
C15	379	TAP2	−6527	32,825,036	32,832,212	7176
C16	7021	HLA-Z	12,989	32,889,395	32,909,479	20,084
C16	4904	lncRNA **	5721	32,889,395	32,909,479	20,084
C17		DMA	3354–13,617	32,956,451	32,966,714	10,263
C17	12,143–1880	BRD2		32,956,451	32,966,714	10,263
C18	1122	DPA1	−5141	33,063,447	33,068,508	5061
C19	−7468	DPB1	59	33,083,458	33,089,755	6297
C20		DPB1	103–1908	33,089,799	33,091,604	1805
C20	1683	DPB2	−122	33,089,799	33,091,604	1805

* negative TE locations start or end within the genes; ** lncRNA is the uncharacterized LOC100294145 (LOC100294145), transcript variant 2, with the USSC ID of ENST00000701517.1, which overlaps the HLA-Z pseudogene.

**Table 5 genes-15-01185-t005:** Distance (bp) between different adjoining transcribed TE clusters.

Adjoining Clusters	Distance (bp) between C	Av * Distance (bp) within C	Distance (bp) between C	Av * Distance (bp) within C	Nearest Gene Loci to C
	NR_SVA_380		R_SVA_27		
C1–C2	6339	550–579			HLA-DRB9
C2–C3	1918	579–1273			HLA-DRB9
C2–C4	7136	579–799	975	2216–373	HLA-DRB9/HLA-DRB5
C3–C4	125	1273–799			HLA-DRB9/HLA-DRB5
C4–C5	1677	799–674	4143	373–698	HLA-DRB9/HLA-DRB5
C5–C6	8726	674–997	894	698–597	HLA-DRB9/HLA-DRB5
C6–C7	2608	997–782	7537	597–1018	HLA-DRB9/HLA-DRB5
C7–C8	7150	782–1224	3959	1018–601	HLA-DRB5/HLA-DRB6
C8–C9	8716	1224–758	7327	601–936	HLA-DRB5/HLA-DRB6
C9–C10	2602	758–2181	2602	936–1996	HLA-DRB6/HLA-DRB1
C10–C11			28,547	1996–1008	HLA-DRB6/HLA-DQA1
C10–C12	38,289	2181–1229			HLA-DRB6/HLA-DQA1
C11–C12			1147	1008–808	HLA-DRB6/HLA-DQA1
C12–C13	155,049	1229–440	4601	808–3582	HLA-DQA1/HLA-DOB
C13–C14	287	440–550			HLA-DOB
C14–C15	5756	550–1435			HLA-DOB/TAP2
C15–C16	57,183	1435–3347			TAP2/LOC10029414
	**R_SVA_85**		**R_SVA_381**		
C2–C7			39,892	311–1877	HLA-DRB9/HLA-DRB5
C7–C9	22,201	1341–1960	22,201	1877–268	HLA-DRB5/HLA-DRB6
C9–C10	12,037	1960–311	3459	268–5313	HLA-DRB6/HLA-DRB1
C10–C17	388,671	311–641	385,273	5313–684	HLA-DRB1/HLA-DMA/BRD2
C17–C18	96,733	641–1265	96,733	684–1265	HLA-DMA/HLA-DPA1
C18–C19	14,950	1265–2099	14,950	1265–1018	HLA-DPA1/HLA-DPB1
C19–C20	44	2099–361	44	2099–361	HLA-DPB1/HLA-DPA2

* Av is average.

**Table 6 genes-15-01185-t006:** The number, percentage, average length (bp) and classifications (class, family, name) of transcribed TEs that are regulated by SVA in the MHC class II region of PPMI cohort.

Class (%) (n, 235)	Family (%)	Name	No. TEs	Length Av * (bp) Data	No. Clusters
DNA (12%)			28	201	9
	hAT (25%)	MER53	7		4
	hAT-Charlie (29%)		8	396	4
		Charlie1,4,14	5		2
		MER1,5,20	3		2
	hAT-Tip100 (7%)	Arthur, MamRep	2	142	2
	TcMar-Mariner (14%)	MADE	4	77	2
	TcMar-Tigger (25%)		7	192	3
		MER2	3		1
		Tigger4,13	4		3
LINE (37%)			87	455	18
	L1 (77%)		67	491	14
	L2 (23%)		20	332	9
LTR (23%)			53	516	14
	ERV1 (43%)		23		
		HERV9	7	735	1
		LOR1	3	465	2
		LTR12	5	694	2
		LTR43	2	267	1
		MER51	2	169	2
		MER52	1	271	1
		MER68	1	492	1
		MER90	2	297	1
	ERVK (15%)		8	1148	6
		HERVK3	3	2186	2
		LTR3	2		2
		LTR14	1		1
		MER3	1		1
		MER11	1		1
	ERVL (6%)		3	308	
		LTR16	1		1
		LTR42	2		2
	ERVL-MaLR (36%)		19	263	10
		MLT	11		6
		MST	6		6
		THE	2		2
SINE (28%)			66	265	18
	Alu (73%)		48	297	15
	MIR (27%)		18	178	10
Retroposon (1%)		SVA	1	1467	1

* Av is average.

**Table 7 genes-15-01185-t007:** TE fragments (n, 16) greater than 1 kb.

Cluster	geneID_TXID	Width bp	repFamily	repClass	chr6 Start	chr6 End	Strand	SVA Regulator (β Effect)			
								380	27	85	381
C3	HERVK3-int_dup96	1836	ERVK	LTR	32,475,495	32,477,330	-	6.93			
C3	HERVK3-int_dup97	1513	ERVK	LTR	32,477,631	32,479,143	-	3.74			
C6	LTR12D_dup185	1013	ERV1	LTR	32,494,693	32,495,705	-		3.09		
C6	HERV9N-int_dup90	2734	ERV1	LTR	32,495,706	32,498,439	+		8.29		
C6	L1MD2_dup3514	1799	L1	LINE	32,501,095	32,502,893	-	3.25	5.54		
C6	L1MD2_dup3396	1545	L1	LINE	32,504,764	32,506,308	-	3.95	8.95		
C6	L1PA6_dup2345	2581	L1	LINE	32,507,704	32,510,284	+	−15.41	48.30		
C7	SVA_B_dup264	1467		Retroposon	32,531,533	32,532,999	+		4.09		
C9	LTR12C_dup1127	1349	ERV1	LTR	32,547,139	32,548,487	-		3.09		
C10	HERVK3-int_dup98	3210	ERVK	LTR	32,560,088	32,563,297	-		2.29		
C10	L2a_dup64990	1085	L2	LINE	32,570,094	32,571,178	+	−4.62	−4.76		3.67
C12a	L1PA13_dup3195	1227	L1	LINE	32,630,000	32,631,226	-	−2.59	3.84		
C12b	L1PA6_dup2346	2584	L1	LINE	32,648,261	32,650,844	+	27.30	56.98		
C12b	MER11C_dup315	1069	ERVK	LTR	32,655,747	32,656,815	-		68.70		
C16	Charlie1_dup741	1909	hAT-Charlie	DNA	32,905,809	32,907,717	-	−3.65			
C17	L1MA4_dup3680	2033	L1	LINE	32,961,235	32,963,267	+			−21.20	19.96

## Data Availability

Data used in the preparation of this article were obtained from the Parkinson’s Progression Markers Initiative (PPMI) database [https://www.ppmi-info.org/, accessed 9 September 2024]. The original contributions presented in the study are included in the article/Appendix A. Further inquiries can be directed to the corresponding author.

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
