# Peer review of "SVA Regulation of Transposable Element Clustered Transcription within the Major Histocompatibility Complex Genomic Class II Region of the Parkinson’s Progression Markers Initiative"

_genes, 2024, doi:10.3390/genes15091185_

Round 1

Reviewer 1 Report

Comments and Suggestions for Authors

The manuscript reports differential TE expression by SVA around MHC class II genes in the publicly available database in the Parkinson’s Progression Markers Initiative. The analyses are complete and through. Although I find that the absence of an explanation for the authors' study's purpose is this manuscript's weakest point. In the abstract (Lines 18-19), the authors wrote, “Our aim was to evaluate …. within chromosome 6.” Their aim is clear, but it's unclear why they chose to examine MHC class II. Is there any association between Parkinson’s disease and MHC class II? These connections were never established in the entire manuscript. It appears from the discussion that the prior study discovered some connections (e.g., ref 35). The reason should have been stated clearly in the abstract and introduction. 

1-    Abstract, Lines 18-19: As I mentioned above, it would be clearer if you could add a purpose or reasoning here. 

2-    Introduction: The authors need to connect MHC class II, Parkinson’s disease, and SVA (see above comments). 

3-    Lines 77-81: While MHC class II molecules are vital for antigen presentation for the extracellular pathogens, TAP and psmb work with MHC class I and are thus vital for intracellular pathogens. Please rephrase it.

4-    Line 92: Please briefly explain what was in reference 35 (see above comment). 

5-    Line 157 “duplicated genes”: TAP1 and TAP2 may have been recently duplicated, but psmb8 and 9 are not duplicates. Archaea already has separate precursor genes for each proteasome gene. Remove “duplicate”.  

Author Response

Comments 1: The manuscript reports differential TE expression by SVA around MHC class II genes in the publicly available database in the Parkinson’s Progression Markers Initiative. The analyses are complete and through. Although I find that the absence of an explanation for the authors' study's purpose is this manuscript's weakest point. In the abstract (Lines 18-19), the authors wrote, “Our aim was to evaluate …. within chromosome 6.” Their aim is clear, but it's unclear why they chose to examine MHC class II. Is there any association between Parkinson’s disease and MHC class II? These connections were never established in the entire manuscript. It appears from the discussion that the prior study discovered some connections (e.g., ref 35). The reason should have been stated clearly in the abstract and introduction.

Response 1:  Thank you for your positive comments, questions and valuable suggestions.

Comments 2: Abstract, Lines 18-19: As I mentioned above, it would be clearer if you could add a purpose or reasoning here.

Response 2: We have included the following sentence in the abstract: ‘Previously, we reported that some expressed SVAs and Human Leucocyte Antigen (HLA) class II genes on chromosome 6 were associated significantly with Parkinson’s disease. Here, our aim was to follow-up our previous study and evaluate the SVA associations and their regulatory effects on the transcription of TEs within the HLA class II genomic region.’

Comments 3: Introduction: The authors need to connect MHC class II, Parkinson’s disease, and SVA (see above comments).

Response 3: We have connected MHC class II, Parkinson’s disease, and SVA as follows: ‘Some of these expressed SVAs and HLA class II gene alleles have been associated with Parkinson’s disease [35].’

Comments 4: - Lines 77-81: While MHC class II molecules are vital for antigen presentation for the extracellular pathogens, TAP and psmb work with MHC class I and are thus vital for intracellular pathogens. Please rephrase it.

Response 4: Thank you for this correction. We deleted ‘intracellular’.

Comments 5: 4- Line 92: Please briefly explain what was in reference 35 (see above comment).

Response 5: We have added the following wrt reference 35: ‘Some of these expressed SVAs and HLA class II gene alleles have been associated with Parkinson’s disease [35]. However, the most significant allelic differences between Parkinson’s disease (PD) and healthy cases after Bonferroni corrections were detected only for the expressed HLA-DRA*01:01:01 and -DQA1*03:01:01 alleles and the NR_SVA_381 genotype. The SVAs that regulated HLA gene transcriptional activities also were allele and haplotype dependent [35]. For example, of the 194 DRB1*15:01/SVA_27 haplotypes in PPMI, 178 (91.8%) were linked to DQA1*01:02/DQB1*06:02.’

Comments 6: 5- Line 157 “duplicated genes”: TAP1 and TAP2 may have been recently duplicated, but psmb8 and 9 are not duplicates. Archaea already has separate precursor genes for each proteasome gene. Remove “duplicate”.

Response 6: Thank you for this correction. ‘Duplicate’ has been removed.

Response 7: For greater clarity about our aims, we have added the following two sentences at the beginning of the last paragraph in the Introduction: ‘Since TEs can influence immune responses [24-27], we hypothesized that the MHC class II SVA genotypes that regulate HLA class II genes might also coregulate the transcription of particular TE families in the MHC class II region, What TE families and loci, if any, are co-expressed with HLA genes in the MHC class II region have not been investigated previously.’

Reviewer 2 Report

Comments and Suggestions for Authors

This is a very interesting study exploring the SINE-VNTR-Alu retrotransposons associations with the transcription of Human Leucocyte Antigen class II genes and TEs within chromosome 6.  The authors performed reanalysis of RNA sequences within the blood transcriptome of a PPMI cohort and TE sequences were identified and annotated from the RNA-seq datafiles. The work is well-written and results are solid with several roadmaps demonstrating the quality of data. Some comments could be taken into account for improvement.

1.      A workflow diagram would help the readers to better understand the goal and strategy of study.

2.      It had been reported that the PCA of two values indicated that the ‘statistic’ value was the first principal component that captured >80% of the total variance in the data sets. Which is the biological meaning of ‘statistic’ focused on distinct groups or clusters?

3.   The quality of figures related to genomic loci map of expressed TE clusters could be increased to help some labels to be more readable. It might be better to put some high-resolution zoom-in figures.

Author Response

Comments 1: This is a very interesting study exploring the SINE-VNTRAlu retrotransposons associations with the transcription of Human Leucocyte Antigen class II genes and TEs within chromosome 6. The authors performed reanalysis of RNA sequences within the blood transcriptome of a PPMI cohort and TE sequences were identified and annotated from the RNA-seq datafiles. The work is well-written and results are solid with several roadmaps demonstrating the quality of data. Some comments could be taken into account for improvement.

Response 1: We thank the reviewer for the positive comments, questions and valuable suggestions.

Comments 2: 1. A workflow diagram would help the readers to better understand the goal and strategy of study.

Response 2: The workflow diagram is provided in the graphic abstract. The reviewer possibly did not see this figure because we submitted it online belatedly.

Comments 3: 2. It had been reported that the PCA of two values indicated that the ‘statistic’ value was the first principal component that captured >80% of the total variance in the data sets. Which is the biological meaning of ‘statistic’ focused on distinct groups or clusters?

Response 3: Thank you for your comment and question. We’ve added the following corrected paragraph to our manuscript in regard to your query about the PCA of the two values: ‘The ‘statistic’ combines both the ‘beta’ value and its variance to provide a more comprehensive measure of the reliability and impact of the SVA effects on TE expression, show the main differences between the SVA effects, and to distinguish between the different groups or clusters in our data. In comparison, ‘beta’, which represents the positive and/or negative effect size, captures less variance in the data, and therefore might not adequately separate distinct clusters. This emphasizes the importance of considering both effect size (‘beta’) and significance (‘statistic’) when interpreting the regulatory impact of SVAs on TE expression. Thus, taken together, PC1 versus PC2 reveals that although the regulatory effects of the four SVAs on TE transcription are markedly different, there also is an overlap between the effects of particular pairs of SVAs.’

Comments 4: 3. The quality of figures related to genomic loci map of expressed TE clusters could be increased to help some labels to be more readable. It might be better to put some high-resolution zoom-in figures.

Response 4: We thank the reviewer for this suggestion. We have left our figures as they are for this paper. We have increased the size of the figures across the page to enlarge the details of the genomic loci maps for better viewing. We plan to publish follow-up papers with more readable figures and fewer labelled items. We will look into the methodology of producing high-resolution zoom-in figures for our future publications.